# Outage Performance of Wireless Powered Decode-and-Forward Relaying Networks in Rician Fading

**DOI:** 10.3390/e24060763

**Published:** 2022-05-28

**Authors:** Zhifei Zhang, Peng Dong, Xinlu Tan, Yaping Li, Ke Xiong

**Affiliations:** 1Engineering Research Center of Network Management Technology for High Speed Railway of Ministry of Education, School of Computer and Information Technology, Beijing Jiaotong University, Beijing 100044, China; zhfzhang@bjtu.edu.cn (Z.Z.); liyaping@chinamobile.com (Y.L.); 2SinoRail HongYuan Software Technology Co., Ltd., Beijing 100044, China; dongpeng@sinorail.com

**Keywords:** energy collection, outage probability, Rician channel, circuit energy threshold, decode-and-forward

## Abstract

This paper investigates the outage performance of simultaneous wireless information and power transfer (SWIPT)-enabled relay networks with the decode-and-forward relaying protocol, where the effect of the energy triggering threshold at the relay on the system performance is considered. The closed-form expressions of the system outage probability and throughput are derived in Rician channel fading. Monte Carlo Simulation method is used to verify the accuracy of the derived closed-form expressions. The effects of some system parameters on the system performances are discussed via simulations, which show that the system outage probability increases with the increase of the minimum transmission rate required by the users and also decreases with the increase of the energy conversion efficiency. Besides, the system throughput increaseswith the increment of the transmit power of the source node, as well as the energy conversion efficiency. Additionally, the outage performance of the system with the equal two-hop distance is better than that of the system with unequal two-hop distance.

## 1. Introduction

### 1.1. Background

The rapid development of mobile communication technology has greatly changed the way of people’s life. High speed, low delay, mass device connection, and low power consumption have become the prominent features of the 5th generation mobile networks (5G). The popularity of the 5G network not only provides reliable communication for unmanned driving, telemedicine, virtual reality, smart home, and other applications, but also brings great improvement to the wireless network device, wireless technology, and derivative industry chains.

Meanwhile, 5G commercial makes it possible for realizing the Internet of Everything (IoE). It is estimated that tens of billions of wireless network devices will be connected to 5G networks in various fields by 2025. Thus, the power supply for large numbers of battery-powered wireless devices will become a thorny problem, which may also lead to a series of environmental problems caused by waste batteries. More importantly, replacing the battery for mass wireless network devices will cost too much. For example, when the wireless network devices are deployed in the mountain top, forest, or chemical plant with harmful gas, it is difficult and dangerous to replace the batteries manually.

Since wireless power transfer (WPT) is able to transmit power from an energy source to energy-limited devices wirelessly [1], it is capable of making network devices get rid of the shackles of wires and providing stable energy without frequent battery replacement. So, it provides an efficient solution to the problem of power supply for the low-power wireless device in Internet of things (IoT). There are two main branches of the research of wireless power transfer. One is the simultaneous wireless information and power transfer (SWIPT) [2], i.e., the simultaneous interpreting of energy and information transmitted by sending signals to the wireless network devices, which was proposed in [3]. Later, Ref. [4] proposed two receiver architectures: time switching receiver (TS) and power splitting receiver (PS) [5]. Another branch is the wireless powered communication network (WPCN) [6,7], where energy and information are transmitted separately over orthogonal channel, which was proposed in [8].

In addition to the limited energy, another big problem in IoT is the long-distance transmission of information. When the distance between the sender and the receiver is relatively far, the signal will suffer from serious fading, which leads to the degradation of communication quality. To release this issue, cooperative relay technology [9] was presented to shorten the data transmission distance, expand the network coverage, improve the transmission reliability, and provide a reliable guarantee for the long-distance transmission of information. One popular relaying protocol is the decode-and-forward (DF) protocol, with which the relay first decodes the information from the signal transmitted from the source, and then forwards the information to the destination node.

### 1.2. Related Work

Due to the channel fading, the wireless transmission may suffer server outage. Therefore, a lot of work have discussed the outage performance of WPT power IoT systems. In [10], the system outage probability and reliable throughput of the wireless-powered SWIPT network was investigated in Rayleigh fading with the imperfect channel state information (CSI) and non-linear energy harvesting (EH) model [11]. In [12], the outage probability of SWIPT amplify-and-forward (AF) relay network based on PS receiving architecture was studied in Rayleigh fading, and the impact of the direct link was discussed. In [13], the secrecy outage performance of a cooperative non-orthogonal multiple access (NOMA) network was investigated in Rayleigh fading, where the SWIPT was adopted. In [14], a battery-assisted SWIPT relaying system was studied in Rayleigh fading, which makes the system can work nomally when the collected energy is not sufficient. In [15], the outage probability of SWIPT AF relay network with cooperative spectrum sharing was analyzed in Rayleigh fading. In [16], the outage performance of SWIPT in network-coded two-way relay systems was investigated in Rayleigh fading, where two transmission protocols, i.e., power splitting two-way relay (PS-TWR) and time switching two-way relay (TS-TWR) protocols were considered. In [17], the outage probability of the proposed relay selection optimization algorithm for SWIPT-enabled cooperative networks was derived in Rayleigh fading. In [18], the outage probability of an AF cooperative NOMA model in multi-relay multi-user networks was studied in Rayleigh fading.

Moreover, in [19], the closed-form expressions of outage probability, bit error rate, and throughput of SWIPT enabled AF relay networks was derived in Nakagami-m fading with opportunistic relaying. In [20], a two-hop single relay network with PS receiver was discussed, where the closed-form expression of the system outage probability was derived in Nakagami-m fading. In [21], the system outage probability of a SWIPT based two-way AF relay network was investigated in Nakagami-m fading, where transceiver hardware impairments (HIs) were considered. In [22], the outage performance of two-hop link networks with direct link and multi-relay cooperation was analyzed in Nakagami-m fading. In [23], the secrecy outage probability of a secure dual-hop mixed radio frequency-free space optical (RF-FSO) downlink SWIPT system was analyzed, where the FSO link and all RF links were assumed to experience Gamma-Gamma, independent, and identical Nakagami-m fading, respectively. In [24], the system outage performance on cooperative cognitive radio network (CCRN) was investigated in Rayleigh fading and Nakagami-m fading, where SWIPT was employed to enable the in-band bidirectional device-to-device (D2D) communications. In [25], the outage performance of SWIPT in cognitive radio sensor networks was analyzed by considering a non-linear energy harvester at the relaying sensor nodes in Nakagami-m fading. In [26], the outage performance of a bidirectional PS-TS SWIPT-based cognitive radio network with the non-linear EH circuit was investigated in Nakagami-m fading channels.

It is noticed that the aforementioned works analyzed the outage performance of SWIPT networks in either Rayleigh fading or Nakagami-m fading. As is known, Rayleigh fading could only describe the channel with rich scattering and non-line of sight (NLoS). Although Nakagami-m fading model is more general to describe the channels with both line of sight (LoS) and NLoS components and it can be simplified to be Rayleigh fading or Rician fading by changing the value of m, Rician fading can more accurately describe the fading distribution in the environment with both direct component and scattering component [27]. Therefore, some work began to evaluate the outage performance of WPT-powered IoT in Rician fading. In [28], the outage probability was minimized in a two-hop source-relay-destination network and the efficient usage of a DF relay for SWIPT toward the energy-constrained destination was proposed in Rician fading. In [29], the outage performance of hybrid protocol based AF relay networks with SWIPT was studied over asymmetric fading channels, where the source-relay link and the relay-destination link are subjected to Rician fading and Rayleigh fading, respectively. In [30], the outage probability of the achievable secrecy rate in the presence of multiple eavesdroppers that employ EH and information decoding (ID) was investigated in Rician fading.

### 1.3. Motivations & Contributions

However, in aforementioned works, the energy triggering threshold of the EH mode was not considered. That is, they ideally assumed that the node can normally work with any amount of available energy. In practice, when the energy collected by the EH node does not reach the energy threshold, the circuit of the node may not be successfully activated. In this case, the system cannot work normally and the outage may also occur. To fill the gap, this paper investigates the outage performance of SWIPT-enabled relay networks with the energy triggering threshold in Rician fading. The contributions of the paper are summarized as follows.

We first derive the closed-form expressions of the outage probability and reliable throughput of the system theoretically. Then, we validate our theoretical analysis by simulation and also discusses the effects of the time assignment factor, the total transmit power of the source node, the Rice factor, the minimum required transmission rate of the system, the energy conversion efficiency, and the two-hop distance on the system performances. The simulation results show that the higher the minimum transmission rate required by the system, the higher the system outage probability. Besides, when the total transmit power of the source node is relatively small, the system throughput is smaller. When the total transmit power of the source node is relatively large, the system throughput is larger. In addition, increasing the system energy conversion efficiency reduces the system outage probability and increases the system throughput. Besides, the impact of the two-hop distance on the system performance is also simulated. Furthermore, the system performance with the equal two-hop distance is slightly better than that with the unequal two-hop distance.

### 1.4. Organization

The rest of the paper is organized as follows. In Section 2, the considered system model is presented. Section 3 theoretically derives the closed-form expressions of the outage probability and reliable throughput of the system. Simulation results are presented in Section 4 to validates our derived theoretical results. Finally, Section 5 draws the conclusions.

## 2. System Model

The considered DF SWIPT-enabled relay network is shown in Figure 1. It includes an energy-rich source node *S*, an energy limited relay node *R*, and a destination node *D*. *R* has no available power, so it needs to collect energy from the source node to receive and transmit information in half-duplex mode. Assuming that there is no direct link from *S* to *D* due the barrier, so *S* can only transmit information to *D* via *R*.

*S* first transmits energy and information to *R*, and *R* uses TS (It is the fact that the PS model usually outperforms the TS one [31]. Nevertheless, the TS model is still a very popular SWIPT receiver model, which has been attracting much interest. The reason for us to choose the TS receiver architecture is that compared with the PS model, the TS one is much simpler to be deployed in practice. Therefore, similar to many recent works [32,33], we consider the TS one in this paper.) receiving architecture to receive energy and information. Then, *R* forwards the information to *D*.

For clarity, the time framework of energy collection and information transmission of DF relay protocol is shown in Figure 2. The whole time period is divided into two stages. The first one is the energy collection stage, and the second one is the information transmission stage, where the information transmission stage is further divided into two parts. In the first part, *R* receives information from *s*, and in the second part, *R* transmits information to *D* by DF relaying. ρ is the time division factor, and *T* is the interval of a time period. Assuming that *R* has no energy before starting transmission, and the two channels are quasi-static, independent and Rician Fading distributed.

### 2.1. Energy Collection

In the first stage with that interval of ρT seconds, *S* transmits energy and information to *R*, and *R* receives energy from the RF signal transmitted by *S* with a TS receiver. Then, the energy collected at *R* can be described by
(1)$$E=\eta \sum_{j=1}^{N_{S}}{\left|h_{S_{j}R}\right|}^{2}P_{S}\rho Td_{SR}^{-m}.$$
where η∈(0,1) is the energy conversion efficiency (Although the non-linear model is more practical and accurate, it has been stated that when the distance between the energy transmitter and the receiver is relatively far or the input energy of the EH module at the receiver is relatively small, the EH circuit may still work in the linear region [34]. Therefore, the research based on the linear EH model is still meaningful in many wireless scenarios. Thus, similar to many existing works [34,35,36], the linear EH model is also adopted to analyze the outage performance of the system in this paper.), $N_{S}>=2$ is the number of source node antennas, and $P_{S}$ is the transmit power of each antenna in the source node (we assume that the transmit power of each antenna is the same, i.e., $P_{S}=P_{\mathrm{sum}}/N_{S}$, with $P_{\mathrm{sum}}$ being the total transmit power of the source node), $h_{S_{j}R}\in \left[1,N_{S}\right]$ denote the channel coefficient of the link from each antenna of *S* to *R*, and $d_{SR}$ denotes the distance from *S* to *R*, and *m* is the path loss index.

### 2.2. Information Transmission

The second stage is the information transmission stage, where in the first time interval (1−ρ)T/2, *R* receives information from *S*, and in the rest time interval (1−ρ)T/2, *R* forwards information to *D* by decoding and forwarding. According to Equation (Equation 1), the available power for *R* to forward information to *D* is
(2)$$P_{R}=\frac{E}{(1-\rho )T/2}=\varphi \sum_{j=1}^{N_{S}}P_{S}{\left|h_{S_{j}R}\right|}^{2}d_{SR}^{-m},$$
where φ=2ηρ/(1−ρ). According to Equation (Equation 2), the signal-to-noise-ratio (SNR) of the first-hop link and second-hop link can be calculated by
(3)$$\begin{array}{cc} \gamma_{R} & =\frac{\sum_{j=1}^{N_{S}}P_{S}{\left|h_{S_{j}R}\right|}^{2}d_{SR}^{-m}}{N_{0}}=\gamma_{0}\sum_{j=1}^{N_{S}}{\left|h_{S_{j}R}\right|}^{2}d_{SR}^{-m}\\ \; & =\gamma_{0}Yd_{SR}^{-m}, \end{array}$$
and
(4)$$\begin{array}{cc} \gamma_{D} & =\frac{P_{R}{\left|h_{RD}\right|}^{2}d_{RD}^{-m}}{N_{0}}=\varphi \gamma_{0}{\left|h_{RD}\right|}^{2}\sum_{j=1}^{N_{S}}{\left|h_{S_{j}R}\right|}^{2}d_{SR}^{-m}d_{RD}^{-m}\\ \; & =\varphi \gamma_{0}XYd_{SR}^{-m}d_{RD}^{-m}, \end{array}$$
respectively, where $\gamma_{0}=P_{S}/N_{0}$ is the SINR of each antenna in the source node, and $h_{RD}$ is the channel coefficients from *R* to *D*. $d_{RD}$ is the distance from *R* to destination node *D*, $X={\left|h_{RD}\right|}^{2}$, and $Y=\sum_{j=1}^{N_{S}}{\left|h_{S_{j}R}\right|}^{2}$.

## 3. Outage Probability and Throughput Analysis

This section analyzes the outage probability and throughput of the network described in Section 2. As mentioned previously, when the energy collected by *R* is less than the triggering threshold of circuit $E_{Q}$ (i.e., $E<E_{Q}$), the relay cannot work normally, and in this case, the system will be outage. When *E* is greater than the triggering threshold of the circuit, the relay works normally, but the information rate affected by fading may also be less than the decoding requirement threshold. In this case, the system outage may also occur. Therefore, the system outage probability can be described by
(5)$$P_{\mathrm{out}}^{\left(e2e\right)}=P_{\mathrm{out}}^{\left(e\right)}+P_{\mathrm{suff}}^{\left(e\right)}*P_{\mathrm{out}}^{\left(\mathrm{info}\right)},$$
where $P_{\mathrm{out}}^{\left(e\right)}$ is the probability that the energy collected is less than the circuit threshold $E_{Q}$ at *R*. $P_{\mathrm{suff}}^{\left(e\right)}$ is the probability that *E* is greater than the circuit threshold $E_{Q}$, where $P_{\mathrm{suff}}^{\left(e\right)}=\left(1-P_{\mathrm{out}}^{\left(e\right)}\right)$ is the probability of sufficient relay energy, and $P_{\mathrm{out}}^{\left(\mathrm{info}\right)}$ represents the probability of link interruption when the relay energy is sufficient.

In terms of the energy collection stage described in Section 2, we have that
(6)$$\begin{array}{cc} P_{\mathrm{out}}^{\left(e\right)}\; & =P_{r}\left(E<E_{Q}\right)\\ \; & =P_{r}\left(\eta \sum_{j=1}^{N_{S}}{\left|h_{S_{j}R}\right|}^{2}P_{S}\rho Td_{SR}^{-m}<E_{Q}\right)\\ \; & =P_{r}\left(Y<\frac{E_{Q}}{\eta P_{S}\rho Td_{SR}^{-m}}\right), \end{array}$$
where $Y=\sum_{j=1}^{N_{S}}{\left|h_{S_{j}R}\right|}^{2}$.

According to [37], the cumulative distribution function (CDF) of the channel gain for the link from *S* to *R* in Rice fading can be expressed by
(7)$$F\left(x\right)=1-\sum_{f=0}^{\Theta_{1}}\sum_{g=0}^{f+N_{S}-1}\frac{{\left(N_{S}K\right)}^{f}{\left(\left(1+K\right)\frac{x}{\lambda }\right)}^{g}}{f!g!e^{N_{S}K+\left(1+K\right)\frac{x}{\lambda }}},$$
where *K* is the Rice factor of the link, λ is the mean value of random variable *x*, and $\Theta_{1}$ with a sufficiently large value.

Following Equation (Equation 7), the energy outage probability of *R* can be re-expressed by
(8)$$P_{\mathrm{out}}^{\left(e\right)}=1-\sum_{f=0}^{\Theta_{1}}\sum_{g=0}^{f+N_{S}-1}\frac{{\left(N_{S}K_{SR}\right)}^{f}D^{g}}{f!g!e^{N_{S}K_{SR}+D}},$$
where $D=\left(1+k_{SR}\right)E_{Q})/\left(\lambda_{SR}\eta P_{S}\rho Td_{SR}^{-m}\right)$, $K_{SR}$ is the Rice factor between for *S* and *R*, and $\lambda_{SR}$ is the mean value of random variable *Y*. As a result, one can obtain the probability as
(9)$$P_{\mathrm{suff}}^{\left(e\right)}=1-P_{\mathrm{out}}^{\left(e\right)}=\sum_{f=0}^{\Theta_{1}}\sum_{g=0}^{f+N_{S}-1}\frac{{\left(N_{S}K_{SR}\right)}^{f}D^{g}}{f!g!e^{N_{S}K_{SR}+D}}.$$

Assuming that the minimum required decoding rate of the system is *R*. Half-duplex relaying is adopted, so that when *S* transmits information to *R*, *R* does not send information to *D*. According to Shannon’s theory [38] one has $R=\left(\left(1-\rho \right)/2\right)\log_{2}\left(1+\gamma_{\mathrm{th}}\right)$. Thus, the system end-to-end SNR threshold can be expressed by $\gamma_{\mathrm{th}}=2^{2R/(1-\rho )}-1$. In terms of Equations (Equation 3) and (Equation 4), the information outage probability when *R* has harvested sufficient power is
(10)$$\begin{array}{cc} P_{\mathrm{out}}^{\left(\mathrm{info}\right)}\; & =\min\left\{\gamma_{R},\gamma_{D}\right\}<\gamma_{\mathrm{th}}\\ \; & =P_{r}\left(\gamma_{R}<\gamma_{\mathrm{th}}\right)+P_{r}\left(\gamma_{D}<\gamma_{\mathrm{th}}\right)\;\;\times -P_{r}\left(\gamma_{R}<\gamma_{\mathrm{th}}\right)\times P_{r}\left(\gamma_{D}<\gamma_{\mathrm{th}}\right)\\ \; & =P_{\mathrm{out}\_R}+P_{\mathrm{out}\_D}-P_{\mathrm{out}\_R}\times P_{\mathrm{out}\_D}, \end{array}$$
where $P_{\mathrm{out}\_R}$ is the outage probability of the first hop link, and $P_{\mathrm{out}\_D}$ is the outage probability of the second hop link. According to Equation (Equation 7), we have that
(11)$$\begin{array}{cc} P_{\mathrm{out}\_R}\; & =F_{Y}\left(\frac{\gamma_{\mathrm{th}}}{\gamma_{0}d_{SR}^{-m}}\right)\\ \; & =1-\sum_{f=0}^{\Theta_{1}}\sum_{g=0}^{\Theta_{1}+N_{S}-1}\frac{{\left(N_{S}K_{SR}\right)}^{f}{\left(\Psi /d_{SR}^{-m}\right)}^{g}}{e^{N_{S}K_{SR}+\Psi /d_{SR}^{-m}}f!g!}, \end{array}$$
where $\Psi =\left(1+K_{SR}\right)\gamma_{\mathrm{th}}/\left(\lambda_{SR}\gamma_{0}\right)$.

We can obtain the probability density function (PDF) of *X*, i.e., the PDF of the link channel gain from *R* to *D* [7], as
(12)$$f\left(x\right)=a\sum_{l=0}^{\Theta_{2}}\frac{{\left(bK\right)}^{l}x^{l}e^{-bx}}{{\left(l!\right)}^{2}},$$
where $a=\left(1+K\right)e^{-K}/\lambda$, b=(1+K)/λ, and λ is the mean value of random variable *x*. It is noticed that the theoretical value of $\Theta_{2}$ is infinite, but in practical, a relatively large value can be selected for calculating.

According to Equations (Equation 7) and (Equation 12), the outage probability of the second hop can be given by
(13)$$\begin{array}{cc} \; & P_{\mathrm{out}\_D}=P_{r}\left(\sum_{j=1}^{N_{S}}{\left|h_{S_{S}R}\right|}^{2}<\frac{\gamma_{\mathrm{th}}}{\varphi \gamma_{0}Xd_{SR}^{-m}d_{RD}^{-m}}\right)\\ \; & =1-\int_{0}^{\Theta_{1}}\sum_{f=0}^{\Theta_{1}}\sum_{g=0}^{f+N_{S}-1}\sum_{l=0}^{\Theta_{2}}\frac{a{\left(bK_{RD}\right)}^{l}{\left(N_{S}K_{SR}\right)}^{f}}{f!g!{\left(l!\right)}^{2}e^{N_{S}K_{SR}}\varphi^{g}}\;\;\times {\left(\frac{\Psi }{d_{SR}^{-m}d_{RD}^{-m}}\right)}^{g}x^{l-g}e^{-\frac{\Psi }{\varphi xd_{SR}^{-m}d_{RD}^{-m}}-bx}dx, \end{array}$$
where $K_{RD}$ is the relay *R* rice factor to destination node *D*, and $\lambda_{RD}$ is the mean value of random variable *X*. $a=\left(1+K_{RD}\right)e^{-K_{RD}}/\lambda_{RD},and\;b=\left(1+K_{RD}\right)/\lambda_{RD}$. See Appendix A for detailed derivation.

Using the equation $\int_{0}^{\Theta_{1}}x^{v-1}e^{-\beta /x-\gamma x}=2{\left(\beta /\gamma \right)}^{v/2}K_{v}\left(2\sqrt{\beta \gamma }\right)$ from [39] and ${\left(x+y\right)}^{m}=\sum_{n=0}^{m}\left(m!x^{m-n}y^{n}\right)/\left(\left(m-n\right)!n!\right)$, the outage probability of the second hop link can be re-expressed by
(14)Pout_D=1−2∑f=0Θ1∑g=0f+NS−1∑l=0Θ2abKRDlNSKSRff!g!(l!)2eNSKSRφg×ΨdSR−mdRD−mDgΨbφdSR−mdRD−ml−g+12×Kl−g+12ΨbφdSR−mdRD−m.
with Equations (Equation 5), (Equation 8), (Equation 9), (Equation 10), (Equation 11) and (Equation 14), the system end-to-end outage probability can be given by
(15)$$\begin{array}{cc} P_{\mathrm{out}}^{\left(e2e\right)}\; & =1-\left(\sum_{f=0}^{\Theta_{1}}\sum_{g=0}^{f+N_{S}-1}\frac{{\left(N_{S}K_{SR}\right)}^{f}{\left(\Psi /d_{SR}^{-m}\right)}^{g}}{e^{N_{S}K_{SR}+\Psi /d_{SR}^{-m}}f!g!}\right)\\ \; & \;\;\times \left(\sum_{f=0}^{\Theta_{1}}\sum_{g=0}^{f+N_{S}-1}\frac{{\left(N_{S}K_{SR}\right)}^{f}D^{g}}{f!g!e^{N_{S}K_{SR}+D}}\right)\\ \; & \;\;\times \sum_{f=0}^{\Theta_{1}}\sum_{g=0}^{f+N_{S}-1}\sum_{l=0}^{\Theta_{2}}\frac{2a{\left(bK_{RD}\right)}^{l}{\left(N_{S}K_{SR}\right)}^{f}\Psi^{g}e^{-N_{S}K_{SR}}}{f!g!{\left(l!\right)}^{2}\varphi^{g}{\left(d_{SR}^{-m}d_{RD}^{-m}\right)}^{g}}\\ \; & \;\;\times {\left(\frac{\Psi }{b\varphi d_{SR}^{-m}d_{RD}^{-m}}\right)}^{\frac{1-g+1}{2}}K_{l-g+1}\left(2\sqrt{\frac{\Psi b}{\varphi d_{SR}^{-m}d_{RD}^{-m}}}\right). \end{array}$$

Following Equation (Equation 15), the system throughput can be given by
(16)$$\begin{array}{cc} T_{h}\; & =R\left(1-P_{\mathrm{out}}^{\left(e2e\right)}\right)\left(1-\rho \right)/2 \end{array}$$

## 4. Simulation Results

In this section, we use Monte Carlo Simulation to verify the correctness of our derived closed-form expressions of the system outage probability and throughput. By fixing some experimental parameters, changing the time assignment factor, the total transmit power of the source node, the Rice factor, the minimum required transmission rate of the system, and the energy conversion efficiency, we discuss the effects of various parameters on the outage probability and throughput of the system. In the simulations, the first hop distance $d_{SR}$ and the second hop distance $d_{RD}$ are expressed as $d_{1}$ and $d_{2}$. We simulate two cases, one is with the equal distance, i.e., $d_{1}=5$ m, $d_{2}=5$ m, another is with the unequal distance, i.e., $d_{1}=4$ m, $d_{2}=7$ m. The experimental results show that the two cases of distance have different effects on the system performance under different parameters.

The icon “Ana” devotes the result of theoretical analysis and is represented by discrete points in the figure. The icon “Sim” represents the results obtained by the Monte Carlo Simulation, which is represented by a solid line in the figure. The Monte Carlo method is usually used to randomly generate the channel coefficients to account for the outage probability. For example, we realize $10^{6}$ times and then count the times of system outage to obtain outage probability. If the numerical result derived by our simulations is consistent with the result obtained by the Monte Carlo simulation, it means that the closed solution is accurate.

### 4.1. Simulation Parameters

The parameters adopted in the simulation experiment are summarized in Table 1. η∈(0,1) is the energy conversion efficiency, whose default value is 0.6 (For example, the 2.45 GHz microwave rectifier can achieve energy conversion efficiency much larger than 0.6 in the linear region [40]. Thus, we select energy conversion efficiency as 0.6 to perform our simulations.). $\lambda_{SR}$ and $\lambda_{RD}$ are the mean values of random variable *Y* and random variable *X*, respectively, and we use λ for them for simplicity. *m* is the path loss coefficient with the default value of 2; R∈(0.5,2) is the minimum transmission rate required by the system with the default value of 1 bit/s/Hz. ρ∈(0,1) is the time assignment factor with the default value of 0.4. $P_{\mathrm{sum}}$ is the total transmit power of the source node, which has the range is (20,40) dBm, and the default value is 30 dBm; $K_{SR}$ and $K_{RD}$ are Rise factors of the links the from *S* to *R* and from *R* to *D*, respectively, and we use *K* for them for simplicity. In order to observe the influence of various parameters on the experimental results, the *K* is set to 3. $E_{Q}$ is the threshold value of relay circuit energy with the default value of $5\times 10^{-5}$ J. $N_{S}\in \left(2,4\right)$ is the number of source node antennas with the default value of 2. $N_{0}$ is the noise power, assuming that the noise power at the relay and destination nodes are both −25 dBm.

### 4.2. Simulation Results and Analysis

Figure 3 shows system outage probability versus ρ with different $P_{\mathrm{sum}}$ under different two-hop distance. As shown in Figure 3, when ρ is around 0.4, the system outage probability is the lowest, i.e., the system outage performance is the best. When ρ≥0.9, the probability of system interruption is 1, which means the system is in a paralyzed state at this time. When $P_{\mathrm{sum}}$ is fixed, the system outage probability decreases first and then increases with the increment of ρ. Because when $P_{\mathrm{sum}}$ is relatively small, the relay node has less time to collect energy, and the probability that the energy collected by the relay reaches $E_{Q}$ is also relatively small. In this case, the system outage probability is relatively high. With the increase of ρ, the relay node has more time to collect energy, so the system outage probability decreases. When the ρ is relatively large, the relay node has more time to collect energy, and the time to transmit information is insufficient, resulting in a larger system outage probability. Besides, when ρ is fixed, the system outage probability decreases with the increment of $P_{\mathrm{sum}}$. Because the greater the value of $P_{\mathrm{sum}}$, the more energy the relay can collect, the higher the probability of reaching $E_{Q}$, and the lower the system outage probability. Moreover, the greater the $P_{\mathrm{sum}}$, the greater the value of ρ when the system is paralyzed. Because the larger the $P_{\mathrm{sum}}$, the more likely it is to accurately transmit information within a unit time. In addition, the system outage performance with equal two-hop distance is slightly better than that with unequal two-hop distance. Last but not the least, the degree of matching between the discrete points and the solid line in the figure is relatively high, i.e., the degree of matching between the theoretical analysis and the Monte Carlo Simulation experiment is high, which proves the correctness of our theoretical analysis.

Figure 4 shows system throughput versus ρ with different $P_{\mathrm{sum}}$ under different two-hop distance. It can be found that when $P_{\mathrm{sum}}$ is fixed, and with the increment of ρ, the system throughput first increases, then decreases, and finally becomes 0. When ρ is fixed, the system throughput increases with the increment of $P_{\mathrm{sum}}$. As $P_{\mathrm{sum}}$ increases, ρ becomes smaller when throughput peaks occur. The reason may be that the larger the $P_{\mathrm{sum}}$, the shorter the time for the relay to collect enough energy, and the earlier the peak throughput occurs. Besides, when $P_{\mathrm{sum}}$ is small, the system throughput is higher when using the equal two-hop distance. When $P_{\mathrm{sum}}$ is relatively large, the throughput difference between the two distance cases is exceedingly small. Furthermore, the solid line in the figure matches the discrete points very well, which proves that the theoretical analysis is accurate.

Figure 5 shows system outage probability versus $P_{\mathrm{sum}}$ with different *K* under different two-hop distance. It is seen from Figure 5 that when *K* is fixed, the system outage probability decreases as $P_{\mathrm{sum}}$ increases. The reason may be that the larger the $P_{\mathrm{sum}}$ with the constant noise power, the larger the SNR, which leads to the more collected energy, the higher the probability of reaching $E_{Q}$, and the smaller the system outage probability. Besides, when $P_{\mathrm{sum}}$ is fixed and *K* is increased, the system outage probability decreases, which means the system outage performance is better. The reason may be that the larger the *K*, the more direct signals, and the more energy collected by the relay, which led to the higher probability of reaching $E_{Q}$, and the lower the system outage probability. In addition, the system outage performance with equal two-hop distance is slightly less than that with unequal two-hop distance.

Figure 6 shows system throughput versus $P_{\mathrm{sum}}$ with different *K* under different two-hop distance. It is seen from Figure 6 that when $P_{\mathrm{sum}}$ is greater than 35 dBm, the throughput increase rate of the system slows down and finally stabilizes at about 0.3 bit/s/Hz. Besides, when *K* is fixed, increase $P_{\mathrm{sum}}$, the throughput of the system also increases until it stabilizes. When $P_{\mathrm{sum}}$ is fixed, increase *K*, and the system outage probability decreases. Moreover, when $P_{\mathrm{sum}}$ is small, the system throughput with equal two-hop distance is slightly higher than that with unequal two-hop distance. When $P_{\mathrm{sum}}$ is relatively large, the system throughput under the two distance cases is almost the same. Comparing Figure 5, the changing trend of system outage probability and throughput is opposite. Such observations are consistent with the theoretical results in Equation (Equation 16).

Figure 7 shows system outage probability versus $P_{\mathrm{sum}}$ with different $r_{\min}$ under different two-hop distance. From Figure 7, it can be found that when $r_{\min}$ is fixed, the system outage probability decreases with the increment of $P_{\mathrm{sum}}$. The reason may be that the greater the value of $P_{\mathrm{sum}}$, the higher the probability that the energy collected by the relay reaches $E_{Q}$, which leads to the better the system outage performance. Besides, when $P_{\mathrm{sum}}$ is fixed, the system outage probability increases as $r_{\min}$ increases. In addition, the system outage performance with equal two-hop distance is slightly smaller than that with unequal two-hop distance.

Figure 8 shows system throughput versus $P_{\mathrm{sum}}$ with different $r_{\min}$ under different two-hop distance. From Figure 8, it can be found that the peak system throughput increases with the increment of $r_{\min}$. Besides, when $P_{\mathrm{sum}}$ is small, the system throughput decreases with the increment of $r_{\min}$. The reason may be that the system outage probability becomes larger in this case, which reduces the system throughput. When $P_{\mathrm{sum}}$ is relatively large, the system throughput increases with the increment of $r_{\min}$. In this case, the system outage probability is small, so $P_{\mathrm{sum}}$ has a minor impact on throughput, while $r_{\min}$ has a significant impact on throughput. Furthermore, the solid line in the figure matches the discrete points very well, which proves that the theoretical analysis is accurate.

Figure 9 shows system outage probability versus η with different $P_{\mathrm{sum}}$ under different two-hop distance. From Figure 9, when $P_{\mathrm{sum}}$ is fixed, the system outage probability decreases with the increment of η. Because the larger the η, the more sufficient energy is collected by the relay, so the system outage probability is relatively small. Besides, when η is fixed, the system outage probability decreases with the increment of $P_{\mathrm{sum}}$. The reason may be that the greater the value of $P_{\mathrm{sum}}$ with the constant noise power, the greater the value of system SNR, which makes the probability that the energy collected by the relay reaches $E_{Q}$ becoming higher, and finally makes the system outage performance is better. In addition, the system outage performance with equal two-hop distance is smaller than that with unequal two-hop distance. Furthermore, the solid line in the figure matches the discrete points very well, which proves that the theoretical analysis is accurate.

Figure 10 shows system throughput versus η with different $P_{\mathrm{sum}}$ under different two-hop distance. According to Figure 10, when $P_{\mathrm{sum}}$ is fixed, increase the system η, the system throughput increases first and eventually stabilize. The reason may be that when η is relatively small, increase the system η, the energy collected by the relay increases, which makes the system outage probability being smaller, and the throughput is greater. When η is relatively large, the energy collected by the relay is enough. At this time, increasing η has little effect on system performance, so the throughput eventually stabilizes. Besides, when the system η is fixed, if $P_{\mathrm{sum}}$ is increased, the throughput also increases. When $P_{\mathrm{sum}}$ is small, the system throughput with equal two-hop distance is slightly larger than that with unequal two-hop distance. When $P_{\mathrm{sum}}$ is relatively large, the throughput difference between the two distance cases is small. Comparing Figure 9, it can be seen that the system outage probability is opposite to the trend of throughput, and the consistent conclusion can also be drawn according to the theoretical result in Equation (Equation 16).

Moreover, we also launch some simulations based on the piece-wise non-linear EH model to compare the difference between the linear EH model and the piece-wise non-linear EH model. Specifically, when the received energy is less than the circuit saturation threshold, we simulate it with the linear EH mode and when the received energy is greater than the circuit saturation threshold, the available energy is set to be the circuit saturation power, which is set to 1 W. To distinguish from the previous simulation results, we set the first hop distance as $d_{1}=3$ m, and the second hop distance as $d_{2}=6$ m. Figure 11 and Figure 12 shows system outage probability and system throughput versus ρ with different $P_{\mathrm{sum}}$ under linear EH model and non-linear EH model. It can be seen that since the system works in the linear region, the simulation results of the linear EH model are consistent with those of the non-linear EH model. The result shows that in our experiments, when the two-hop distances are $d_{1}=3$ m and $d_{2}=6$ m, the system works in the linear region and does not enter saturation. This is consistent with many existing works [34,35]. When the distance between the energy transmitter and the receiver is relatively far or the input energy of the EH module at the receiver is relatively small, the EH circuit may still work in the linear region.

## 5. Conclusions

This paper investigated the outage performance of SWIPT-enabled relay networks with the DF relaying protocol, where the effect of the energy triggering threshold at the relay on the system performance was considered. The closed-form expressions of the system outage probability and throughput were derived in Rician channel fading. The Monte Carlo Simulation method was used to verify the accuracy of the derived closed-form expressions. The effects of some system parameters were discussed via simulations. It has been observed that the system outage probability increase with the increase of the minimum transmission rate required by the system and decreases with the increase of the energy conversion efficiency. Besides, with increased source node transmit power or the energy conversion efficiency, the system throughput increase and vice versa. In addition, the outage performance of the system with an equal two-hop distance is better than with an unequal two-hop distance.

In [41], three CSI-free schemes were proposed, i.e., one antenna (OA) scheme, all antennas at once (AA) scheme, and switching antennas (SA) scheme. In OA, only one of PB’s antennas transmits with full power. In AA, all antennas of PB transmit at the same time by sharing the available transmit power. In SA, PB transmits with full power by selecting one antenna at each time slot, and all antennas are used during a block. Actually, the scheme used in our work was very similar to the AA scheme, where we focused on evaluate the outage performance of SWIPT-enabled relay networks in Rice fading channel. In the future work, we may consider OA and SA in the SWIPT-enabled relay networks.

## Figures and Tables

**Figure 1 entropy-24-00763-f001:**
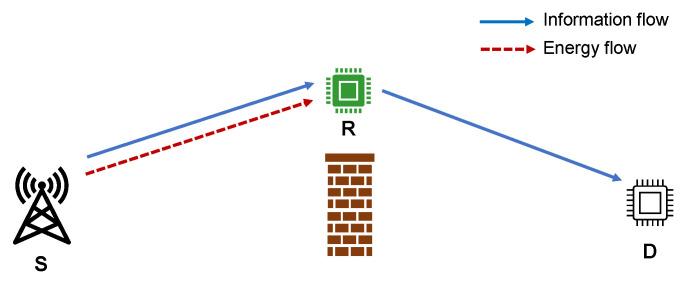
Two-hop relay network model with SWIPT.

**Figure 2 entropy-24-00763-f002:**
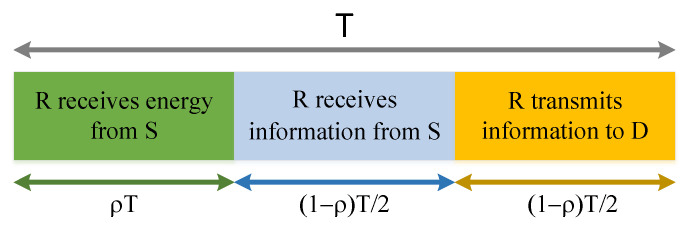
Time frame of the SWIPT-enabled relaying with TS protocol.

**Figure 3 entropy-24-00763-f003:**
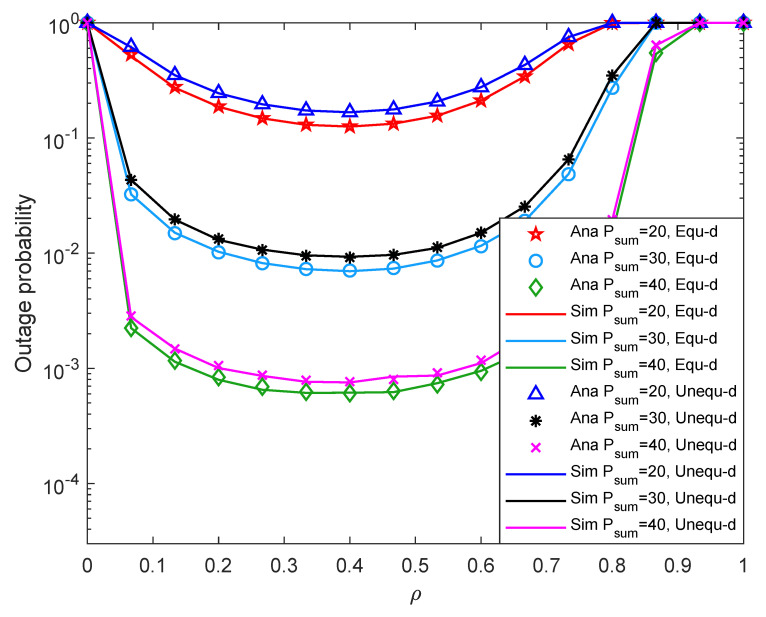
Outage probability versus the time assignment factor.

**Figure 4 entropy-24-00763-f004:**
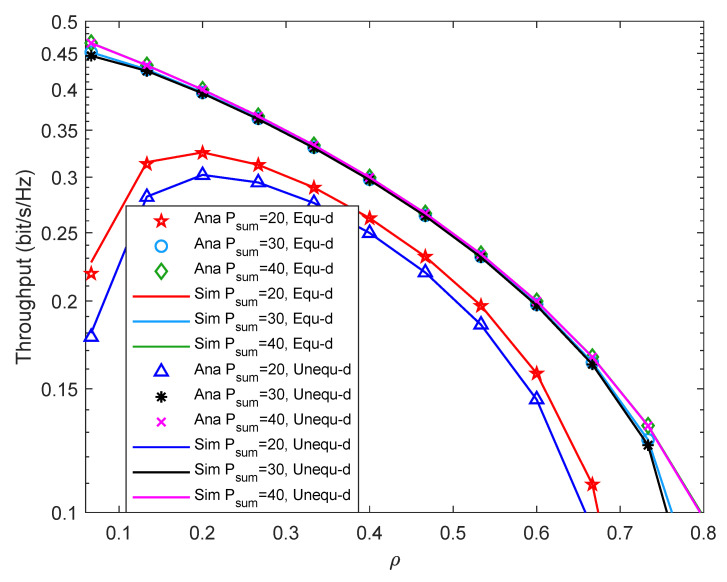
Throughput versus the time assignment factor.

**Figure 5 entropy-24-00763-f005:**
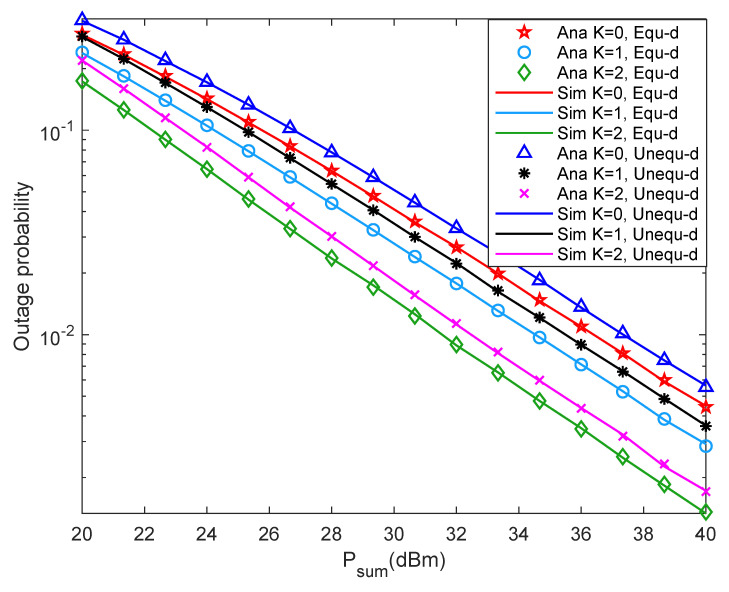
Outage probability versus Rice factor.

**Figure 6 entropy-24-00763-f006:**
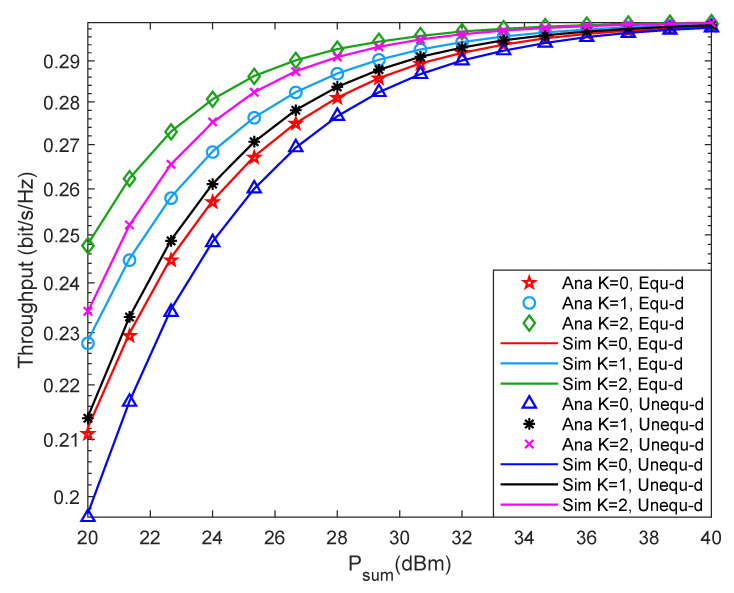
Throughput versus Rice factor.

**Figure 7 entropy-24-00763-f007:**
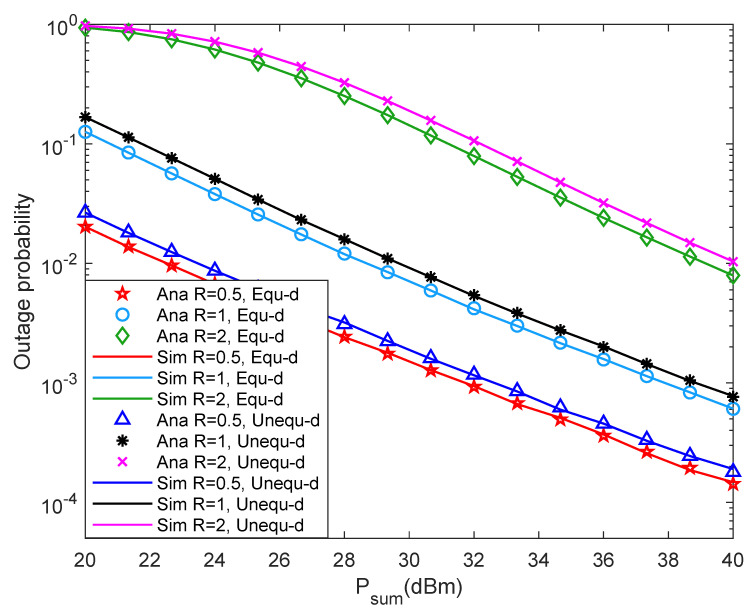
Outage probability versus transmission rate.

**Figure 8 entropy-24-00763-f008:**
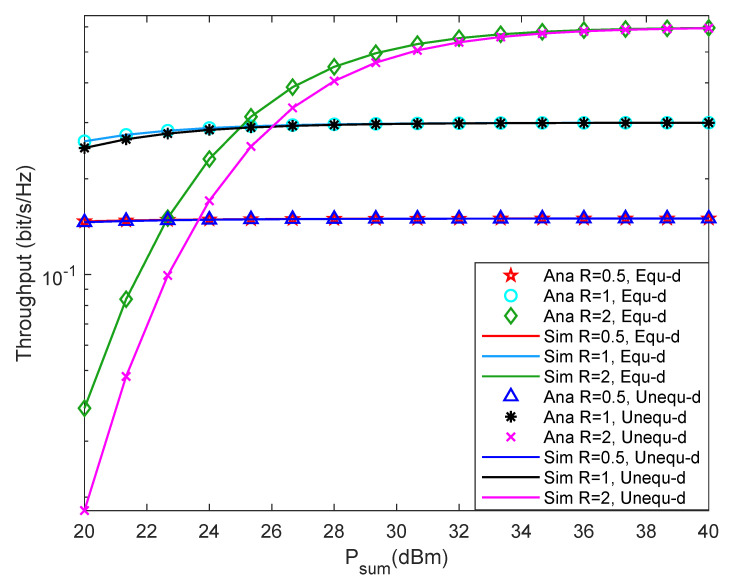
Throughput versus transmission rate.

**Figure 9 entropy-24-00763-f009:**
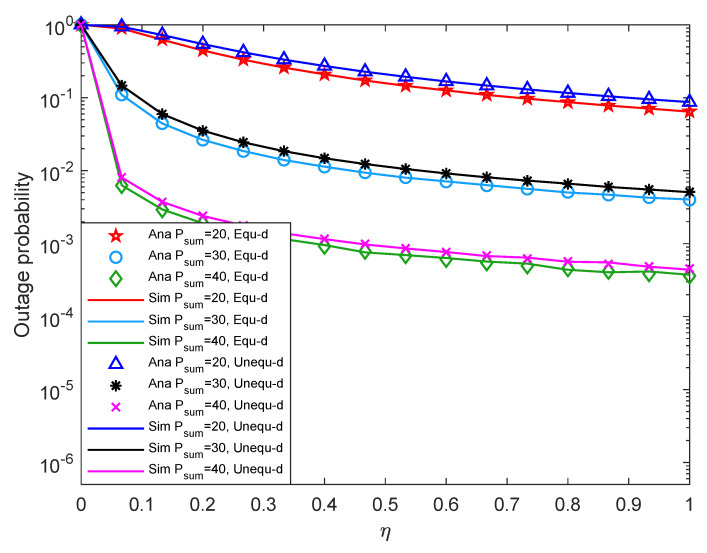
Outage probability versus energy conversion efficiency.

**Figure 10 entropy-24-00763-f010:**
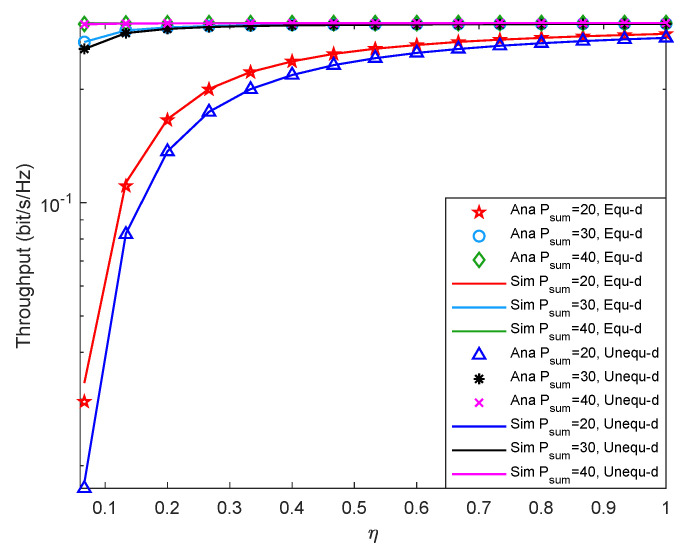
Throughput versus energy conversion efficiency.

**Figure 11 entropy-24-00763-f011:**
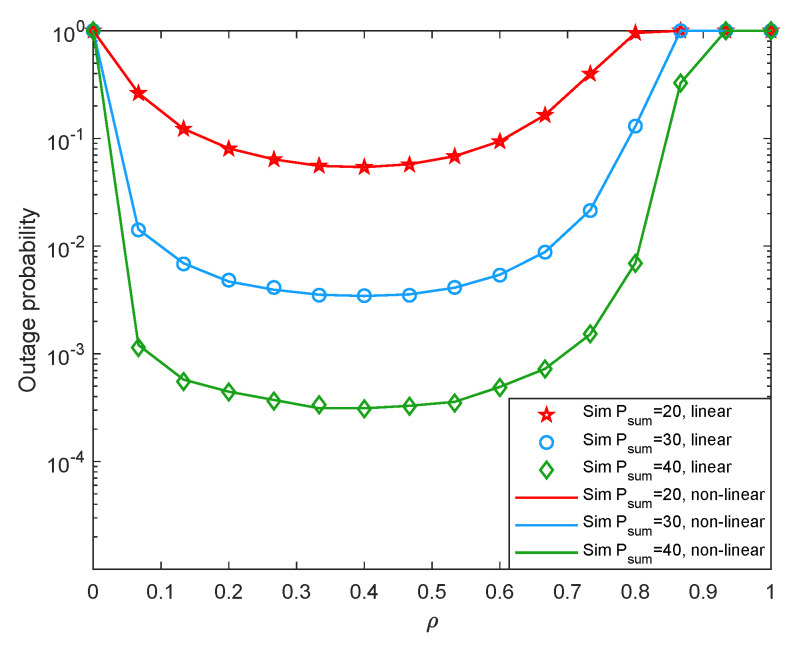
Outage probability versus the time assignment factor under non-linear EH model.

**Figure 12 entropy-24-00763-f012:**
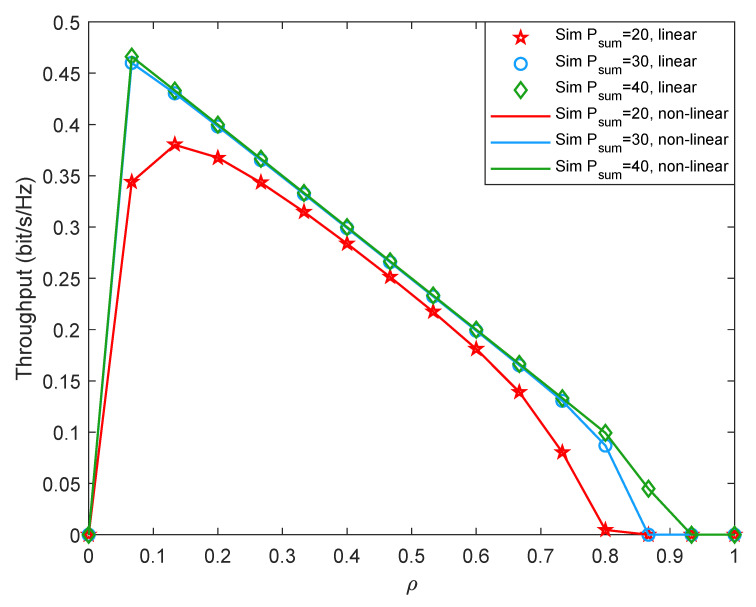
Throughput versus the time assignment factor under non-linear EH model.

**Table 1 entropy-24-00763-t001:** Parameters used in the simulation.

Symbol	Meaning	Value
η	Energy conversion efficiency	0.6
λ	The mean of channel gain	1
*m*	Path loss coefficient	2
$r_{\min}$	Minimum transmission rate required by system	1 bit/s/Hz
ρ	Time assignment factor	0.4
$P_{\mathrm{sum}}$	Total transmission power of source node	30 dBm
*K*	Rice factor	3
$E_{Q}$	Threshold of circuit energy	$5\times 10^{-5}$ J
$N_{S}$	Number of source nodes antenna	2
$N_{0}$	Noise power	−25 dBm

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
