# Peer review of "Outage Performance of Wireless Powered Decode-and-Forward Relaying Networks in Rician Fading"

_entropy, 2022, doi:10.3390/e24060763_

Round 1

Reviewer 1 Report

The paper is very interesting and you can see the effort made to complete it. The following corrections should be made to improve the quality of the paper:

1… The abstract must be modified, adding the most important things found in the investigation, including the most interesting results and conclusions.

2… The problem statement should be improved, showing the weaknesses of previous research and showing the benefits of this research.

3… The problem statement needs to be improved.

4… The introduction should be greatly improved since the objectives of the research are not clear and the contributions of the research are not seen either

5… The conclusions must be improved, which must be consistent with the results obtained.

Author Response

Thanks for your comments. We summarize our response in the appendix file.

Reviewer 2 Report

This paper analyzes the outage probability and reliable throughput of information and power transfer decode-and-forward system functioning in the communication channel with Rician fading.  A distinguishing feature of this analysis is that the authors take into account the energy triggering threshold, that is, the effect that if collected energy is less than a given threshold the system cannot work normally.  The main contributions of the paper are closed-form expression for the outage probability and reliable throughput.  

In my opinion, this paper can be published after addressing the following remarks:

  1. Section 3, formulas (6)-(16). These derivations are rather difficult to follow. I would recommend explaining the main idea behind them and giving the main steps of the proof. The detailed derivations could be put in the appendix.
  2. 6, before the formula (7). I did not understand authors’ explanation for the form of CDF. A reference is needed.
  3. 6 After (9), “According to Shannon theory, one have that R=….” Should be “ According to Shannon’s theory one has…”, the statement needs a reference.
  4. Section 4, the purpose of the presented comparison of the calculation results and simulations is not clear to me. If formulas (15), and (16) give tight results then they do not require any verification by simulations.  

Author Response

(The authors gave the same response as above.)

Reviewer 3 Report

This work is in general well-written, but the novelty is very limited. Similar investigations have been carried out before, basically changing the channel model. Thus, it is rather unclear what can be learned from this work that could not be inferred from the literature.

Moreover, may other main concerns are as follows:

  • The energy and information transmission methods from the source are not clear. The source is a multi antenna device, but how are these multiple antennas being used? By inspecting the energy and information outage probabilities, it seems that the authors assume coherent combining in the MISO channel between source and relay. Please clarify that, as this would require channel state information (CSI) at the transmitter side.
  • In case CSI is needed at the transmitter, how could this be obtained in a setup where the relay has no other source of energy but that transmitted by the source node?
  • If no-CSI is available, or even if partial CSI is considered, I strongly suggest the authors look at the works of Onel Alcaraz Lopez on the use of multiple antenna power beacons. 
  • The linear RF-DC energy conversion model is not convincing. The authors say that the non-linear model can be well approximated by a linear model when operating close to the sensitivity region. That is ok. But, in such cases the efficiency would be far less than 60%, as such high efficiency values are obtained very close to the saturation region.
  • The parameters used in the numerical results should be clearly justified, specially the energy threshold, also known as the energy harvesting sensitivity. Practical values should be utilized, otherwise the results are meaningless.

Author Response

(The authors gave the same response as above.)

Round 2

Reviewer 3 Report

Thanks for the detailed answers to my previous comments. However, there is one questions that has not been completely answered:

  • The authors mentiond that practical RF-DC conversion circuits can achieve efficiencies even larger than 60%. That is fine, as is the case of the powercast RF-DC converter. However, please note that the RF-DC conversion is highly non-linear. That is clear from page 6, in the Product Datasheet of Powercast P1110B – 915 MHz RF Powerharvester Receiver. Why is this not being considered in this paper? Why are you considering linear RF-DC transfer function if that is not practical?

Author Response

(The authors gave the same response as above.)

Round 3

Reviewer 3 Report

The authors insist in considering linear RF-DC conversion, what is not practical. My comment was related to the fact that conversion in non-linear, but the authors focus their answer on the efficiency. I have already mentioned that the efficiency is fine. The problem is to assume linear RF-DC conversion.

In this response the authors mention a RF-DC converter that operates in a quasi-linear region between 10mw and 25mW, a 4dB long region, what is really small for a fading channel. Morever, it is clear that the RF-DC transfer function saturates (i.e., it is non-linear). The reference used by the authors to justify the linear RF-DC operation actually is quite clear in saying that the saturation effect is very strong (i.e., conversion is non-linear).

I really do not understand why the authors do not present results considering the non-linear conversion. I will not be convinced that you can assume the conversion to be linear. Some previous papers in this area considered linear conversion, but the area has evolved and we must consider more practical models.

Why do not the authors consider a simple non linear model like that in E. Boshkovska, D. W. K. Ng, N. Zlatanov and R. Schober, "Practical Non-Linear Energy Harvesting Model and Resource Allocation for SWIPT Systems," in IEEE Communications Letters, vol. 19, no. 12, pp. 2082-2085, Dec. 2015, doi: 10.1109/LCOMM.2015.2478460?

Nevertheless, I leave the decision to the editor. 

Author Response

(The authors gave the same response as above.)
